ecology, environmental science, evolution

local adaptation, temperature adaptation, life history, freshwater organisms, diversity panel

**Author for correspondence:**
Dieter Ebert
e-mail: dieter.ebert@unibas.ch

# Temperature- versus precipitation-limitation shape local temperature tolerance in a Holarctic freshwater crustacean

Leonie Seefeldt and Dieter Ebert

Department of Environmental Sciences, Zoology, University of Basel, Vesalgasse 1, 4051 Basel, Switzerland

DE, 0000-0003-2653-3772

Species with wide geographical distributions are often adapted locally to the prevailing temperatures. To understand how they respond to ongoing climatic change, we must appreciate the interplay between temperature, seasonality and the organism's life cycle. The temperature experienced by many organisms results from an often-overlooked combination of climate and phenology. Summer-active (high latitude) populations are expected to adapt to local summer temperatures, but this is not expected for populations that outlive the summer in their dormant stage (low latitude, precipitation-limited). We recorded reproduction and survival in genotypes from 123 Holarctic populations of *Daphnia magna* during a multi-generation thermal ramp experiment. Genotypes from summer-active populations showed a positive relationship between heat tolerance and local summer temperature, whereas winter-active populations did not. These findings are consistent with the hypothesis that *D. magna* adapts to the local temperatures the animals experience during their planktonic phase. We conclude that predicting local temperature adaptation, in particular in the light of climate change, needs to consider the phenology of geographically wide-ranging species.

## 1. Introduction

As average and extreme temperatures increase [1], flora and fauna have been predicted to adapt to these new climatic conditions in their local habitats, for example, by evolving tolerance to elevated temperatures. This prediction is consistent with observations that organisms are locally adapted to their prevailing climatic conditions [2–4] and also with reports demonstrating that local populations can adapt to changing climatic conditions over short time periods [5–7]. To predict how populations cope with locally changing temperature conditions, it is necessary to understand both the selective agent (i.e. the local temperature regime as experienced by the organisms) and the ability of a population to respond to local selection (i.e. potential for local adaptation).

Latitude is often considered to be a good proxy for local temperature in studies that consider wide geographical ranges [4,8–10]: as latitude increases, temperature generally decreases. This is, however, not always the case; in fact, the correlation between latitude and experienced temperature has limits [11]. For example, some organisms start their growth period earlier in warmer climates, so the local temperature conditions that trigger the start of the active period (e.g. seed germination, hatching or eclosion from resting stages) may not differ across latitudes [12]. Latitude–temperature mismatch occurs in a more extreme way for populations that are active during different seasons of the year. At higher latitudes, for example, the growing season usually happens during the warmest part of the year, because winter temperatures prevent physiological function. Towards the equator, however, the

growing season may be in the colder part of the year because temperatures are too high and/or precipitation too low during the hottest season. For example, the mosquito *Culex pipiens*—a vector of arboviruses, including West Nile virus—has a very wide geographical distribution [13]. In central and northern Europe, its active stage is found exclusively during the summer, while in subtropical regions, it is mainly found during the wet season—the coldest part of the year. During other seasons, it undergoes diapause. Other insect species, including butterflies, dipterans and dragonflies, exhibit similar patterns [14–17]. For these examples, temperature experienced by the population outside its dormant phase cannot be predicted by latitude alone. Therefore, the phenology of the organism needs to be taken into account.

To understand the link between phenology, diapause and local temperature adaptation, we use the planktonic crustacean *Daphnia magna* which lives in fresh- and brackish-water habitats of the Holarctic with a latitudinal range of about 30–70° north [18]. Because of its wide geographical distribution and the resulting wide range of local temperatures in which it is found, this species is well suited for investigating adaptation to local temperatures. *Daphnia magna* has been shown to respond to increased temperature with an evolutionary increase in their upper temperature tolerance [5]. Since *Daphnia* can only persist in the planktonic phase in temperature ranges of about 5–30°C [19], they can only be found during certain seasons in any given latitude, while they diapause in the form of resting eggs (so-called ephippia) at other times. In Europe, north of about 43°, planktonic *D. magna* are typically found in the warmest season (here called summer-active), while south of this latitude, the crustacean emerges from resting stages only at the beginning of the rainy season, the colder half of the year (winter-active). Thus, winter-active populations are not expected to experience and thus adapt to summer temperatures, as summer-active populations do. Previous studies of temperature adaptation in aquatic invertebrates [20–22] did not consider this variation in phenology, potentially explaining why they found poor support for the local adaptation hypothesis. Four studies testing for local temperature adaptation in *Daphnia* also found mixed evidence, possibly because these studies did not differentiate summer- and winter-active populations in their design [20,23–25].

The aim of this study is to test populations of *D. magna* for local temperature adaptation, taking the latitudinal cline in phenology into account. For this, we tested the temperature tolerance of *D. magna* from 123 populations across the Holarctic (using the *Daphnia magna* Diversity Panel) [26]. One clone per population was kept at the laboratory and tested at temperatures up to 37°C, while we recorded fecundity and survival. As predicted, we found a positive correlation between the maximum temperature tolerance and both latitude and the warmest summer temperatures only for animals from summer-active populations. By contrast, temperature tolerance of animals originating from winter-active populations could not be explained with these environmental temperature proxies. We conclude that only in summer-active populations do summer temperatures limit biological function, thus climate warming may directly influence patterns of local adaptation in these populations.

## 2. Material and methods

### (a) Temperature loggers in natural habitats

To understand the seasonal dynamics of water presence/absence and water temperature across Europe and the Middle East, we placed TidBits dataloggers (Onset Computer Corporation, Bourne, MA, USA) close to the centre of six ponds along a transect from the White Sea (about 66° N) to Israel (about 30° N) and recorded the temperature every 30 min. The datasets cover at least 2 years in each case. Battery failure, vandalism and extreme weather conditions affected the length of time that data could be recorded.

### (b) Assessment of growth season

To assess when the *D. magna* population is in its planktonic phase (summer- or winter-active population), we relied on information from local scientists. Furthermore, we used monthly climatic data (temperature and precipitation) from WorldClim database (www.worldclim.org) for each location and consulted published information about the water body. If more information was needed, we used diverse Internet resources such as photographs and areal pictures (current and historical) from Google Earth and Google Maps to assess the seasonality of the water body. Locations where *D. magna* are found year-round were recorded as summer-active, as we were interested in the upper temperature tolerance of the animals.

### (c) Animals and culture conditions

We used 123 *D. magna* clones collected from the Holarctic climate zone (i.e. Europe, Asia, North Africa and North America; electronic supplementary material, table S1). These clones came from the *Daphnia magna* Diversity Panel that has been used to study geographical variation among *D. magna* genotypes [25–28]. Each genotype was collected from a different population and was kept clonally (iso-female lines) in standard culture conditions: 20°C, a light : dark cycle of 16 : 8, artificial *Daphnia* medium (ADaM), and a 1 : 1 (by cell count) mixture of *Nanochloropsis limnetica* and *Acutodesmus obliquus* algae as the only food (50 million cells per 380 ml jar three times a week).

### (d) Assay of temperature tolerance

Animals were taken from the stock collection (at 20°C) and cultured in 380 ml jars filled with 350 ml *Daphnia* medium at 23°C. Each jar was stocked with five females and we produced five such replicate populations in 380 ml jars and kept them in one incubator. All replicate jars were assigned random numbers, to keep the identity of the clones anonymous during their handling. The medium was changed once a week, and males were removed to allow better conditions for population growth. During the acclimation phase, every two weeks, the temperature was raised (23, 25, 26, 27 and 28°C), and population survival and the presence of juveniles (as an indicator of successful reproduction) were recorded. All replicate populations that were able to reproduce were then kept at 28°C. We then took animals from these 28°C cultures and tested for reproduction and survival at nine higher temperatures (29–37°C in increments of 1°C; see electronic supplementary material, figure S1, for an overview). For each of these higher temperatures, we collected three juvenile females (1–3 days old) from each of the five replicate populations kept at 28°C and placed them in a 100 ml jar with 80 ml medium. These jars were then placed in a walk-in climate chamber at the higher temperature (29–37°C). Every second day for the following 14 days, the animals were checked for survival and signs of reproduction (eggs in brood pouch and newborns in the medium). The limited availability of climate chambers did not allow us to test more than one higher temperature at a time. If

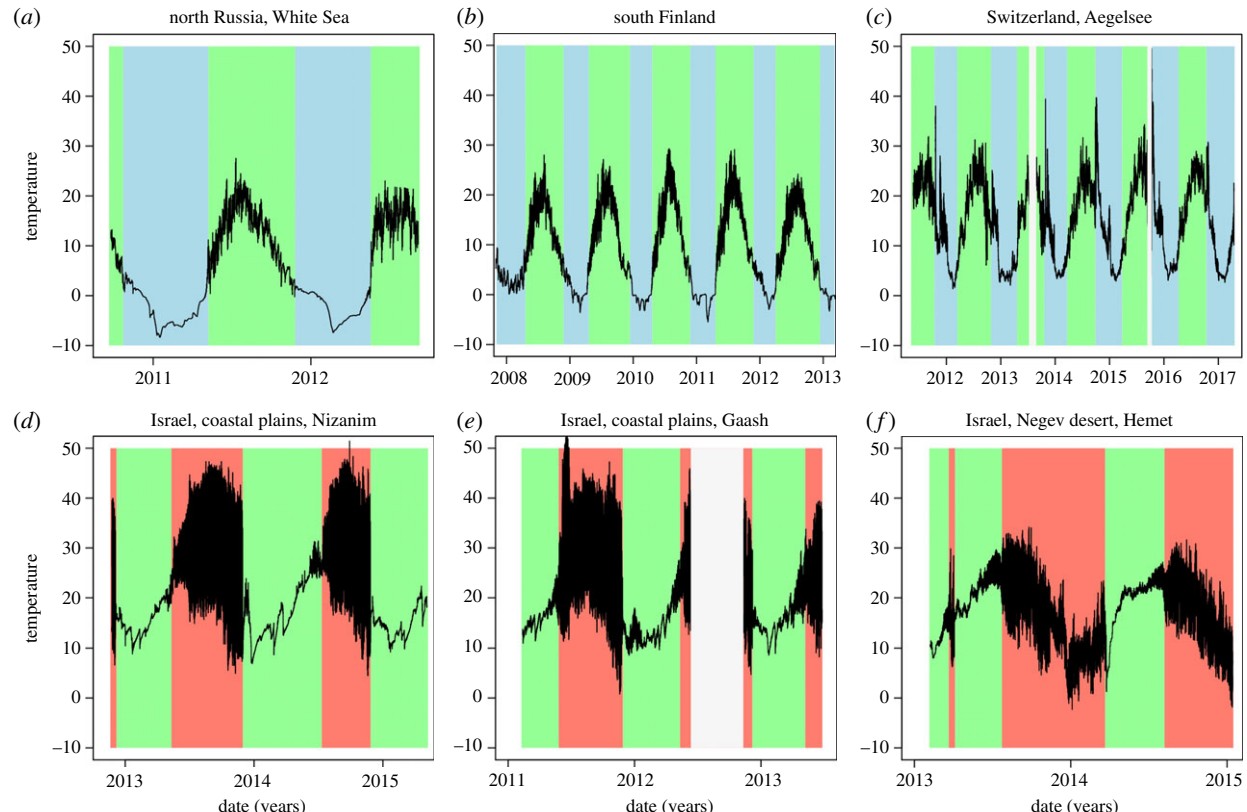

**Figure 1.** Seasonal temperature dynamics of water bodies from three summer-active northern populations (*a–c*) and three winter-active southern populations (*d–f*). The black line marks temperature recorded in 30 min intervals. Time periods with water conditions suitable for planktonic *Daphnia* have a green background; time periods with unsuitable conditions have a blue (less than 5°C, northern populations) or a red background (dry, southern populations). During dry periods, daily temperature fluctuations are much stronger. In northern water bodies, suitable periods are at the hottest period of the year, while in the south, suitable periods fall in the rainy winter season. Data logger failure (low battery and vandalism) is shown in white. The temperature peaks in late autumn (October/November) in the Aegelsee pond of Switzerland are caused by hot water inflow (condensation water) from a sugar factory (the pond is used as a sewage pond from October to December). The lower summer temperatures of the desert population (*f*) are caused by vegetation shading the data logger. (Online version in colour.)

at least one animal of a replicate survived the 14-day test period, the same replicate was tested at the next higher temperature, but starting with juveniles taken from the 28°C cultures. If none of the animals survived 14 days at a certain temperature, this replicate was still tested at the next higher temperature. If all animals died again, we assumed that we had exceeded its upper temperature tolerance and discontinued with this replicate. As four replicates of one clone survived 35°C, we stopped the experiment after testing these replicates at 36 and 37°C.

## (e) Data handling and statistical analysis

All data handling and analysis was done with the R software v. 3.4.1. For each replicate, we calculated the maximum survival temperature as the average between the highest temperature at which the animal was able to survive for two weeks and the lowest temperature at which it was not. Likewise, the maximum temperature for reproduction was calculated as the average of the highest temperature at which successful reproduction was observed (production of free-swimming offspring within a two-week period) and the lowest temperature at which no free-swimming offspring was observed.

Temperature data (monthly averages from years 1970 to 2000) were downloaded from the Worldclim Database for each of our sample locations ( electronic supplementary material, table S1). Locations within 1 km to the sea had to be shifted about 500 m inland to meet the requirements of the database (there are no data for marine sites). Following Yampolsky *et al.* [25], we calculated the mean of the highest temperature (tmax in WorldClim, in all cases either July or August) as an

estimate of the average high temperature of the warmest month (AHT_warmest) for all locations.

Genetic variance components for the clones were calculated with VarCorr of R package nlme (VarCorr(lme(traits ~ clone, data = dat, random = ~1 | clone)). We used analysis of covariance to test for the effect of latitude, average high temperature of the warmest month (AHT_warmest) and the heterogeneity of slopes for summer- and winter-active populations (lm(trait ~ summer_active × covariable)) (with summer_active being either 0 or 1 and covariable being either latitude or the average high temperature of the warmest month (AHT_warmest)). A significant interaction term indicates a lack of homogeneity of slopes.

For the contour maps and the spatial autocorrelation analysis, we used clonal means for each trait. For the contour maps (R package fields), a thin plate spline surface was fitted to our irregularly spaced data, with the smoothing parameter determined by generalized cross-validation. These fitted surfaces were superimposed on a map of the Western Palaearctic (data points between −10° and 60°), where we had a sufficiently dense sampling grid to make this approach meaningful. We used the 'plasma' colour scheme of the R package viridis to better represent the results on a grey scale and to make it easier to interpret them by those with colour blindness.

For the spatial autocorrelation, the distance intervals among sampling sites were chosen to be 200 km up until 2000 km; from there, we used intervals of 2000 km up until 8000 km. Larger distances were pooled in one distance class. This method allowed each distance class to have approximately the same number of data points. For the autocorrelation analysis (calculating Moran's *I*), we used the function correlog in the R package ncf [29].

# 3. Results

## (a) Seasonality and habitat suitability

The data from all temperature loggers reveal a clear seasonality with high mid-year, summer temperatures and low winter temperatures (figure 1). The three locations in Israel show high fluctuations (black area) during the warmest part of the season, but show less daily variation during the colder period. The high daily fluctuations indicate times when these ponds were dry and the data-loggers recorded fluctuations in air temperature rather than water temperature. The coldest points of the year at these ponds (i.e. spring and winter, when the temperature is generally below 20°C) are when *Daphnia* typically find suitable habitats (figure 1d–f, green background). In northern populations (north Russia, southern Finland, Switzerland; figure 1a–c), *Daphnia* find suitable conditions, only during spring, summer and autumn when temperatures are above about 5°C.

The assessment of habitat seasonality for the populations showed a clear distinction between north and south. Summer-active populations are found mostly north of 43° N, when average high temperatures of the warmest month are below 28°C (figure 2). The spatial distribution of winter-active population corresponds very well with the distribution of the subtropical climate zone following the Köppen classification [30]. There is a wide transition zone between summer- and winter-active populations found between 35° and 55° N, because the habitat classification is not only dependent on latitude, but also on other factors, such as altitude and local climate.

## (b) *Daphnia* upper temperature tolerance

The maximal survival temperature for the tested *D. magna* clones ranged between 28 and 35°C, whereas the maximal reproduction temperature ranged between 26 and 32°C. Clonal means for maximal reproduction temperature and maximal survival temperature are strongly correlated (Pearson $r = 0.84$, $p < 0.0001$). This variation among clones can be largely attributed to genetic effects: clone effects explained 58.72% of the total phenotypic variation for the maximal reproduction temperature (mean$^2$ = 5.243, $F_{122,496} = 15.77$, $p < 0.0001$) and 58.73% of the total variation (mean$^2$ = 9.032, $F_{122,496} = 22.35$, $p < 0.0001$) for the maximal survival temperature.

As predicted, temperature tolerance increased with increasing average high temperature of the warmest month and decreasing latitude only in the genotypes from the summer-active populations (figure 3 and table 1). Genotypes from winter-active (subtropical) locations were highly variable in their tolerance and showed no correlation with average high temperature of the warmest month and latitude (figure 3). There were no significant correlations between the winter-active population's maximal temperatures for survival and fecundity and different winter temperature estimates (average temperature in March, average temperature of the coldest quarter, average temperature of the wettest quarter) (Pearson's correlation: $-0.1 < r < 0.1$; $p > 0.2$ in all cases).

The nonlinear distribution of temperature tolerance across the Western Palaearctic is also seen when the smoothed temperature tolerance estimates are superimposed onto a map of the Western Palaearctic (between −10° and 60° E;

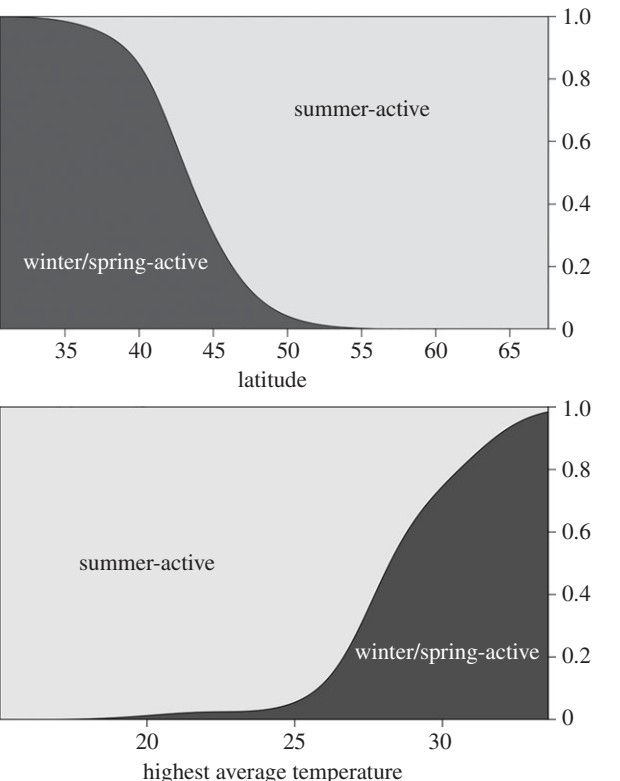

**Figure 2.** Density plots for winter-active and summer-active populations plotted against latitude and against the average temperature of the warmest month.

our sampling density was too low outside this geographical region to make this approach meaningful there) (figure 4). While summer-active populations (roughly north of 43° N) show a clear north–south cline, such a cline is not recognizable for the winter-active populations.

An analysis of spatial autocorrelation of temperature tolerance also supported the finding that *D. magna* are adapted to local temperature conditions. Strong positive autocorrelations for temperature tolerance are seen on distances up to about 1000 km (figure 4), indicating that clones collected close to each other show a similar upper temperature tolerance. Since local climate changes gradually across space, such a pattern is expected for adaptation to climate variables.

# 4. Discussion

For species whose geographical distribution is limited by different factors across their range, latitude may be a bad proxy for expected local temperature adaptation. Freshwater organisms in small water bodies are a good test case for this: in part of their range, their habitat can be limited by the presence of water, while in other parts, high temperatures in mid-summer can limit their active phase. Adverse conditions are outlived in form of resting stages. Therefore, local adaptation to high summer temperatures is only expected in habitats with active populations during summer. We tested this prediction by investigating the upper limit of temperature tolerance for survival and reproduction of *D. magna* genotypes collected across large parts of the Holarctic climate zone. Unlike earlier studies, we accounted for the fact that northern populations experience the highest water

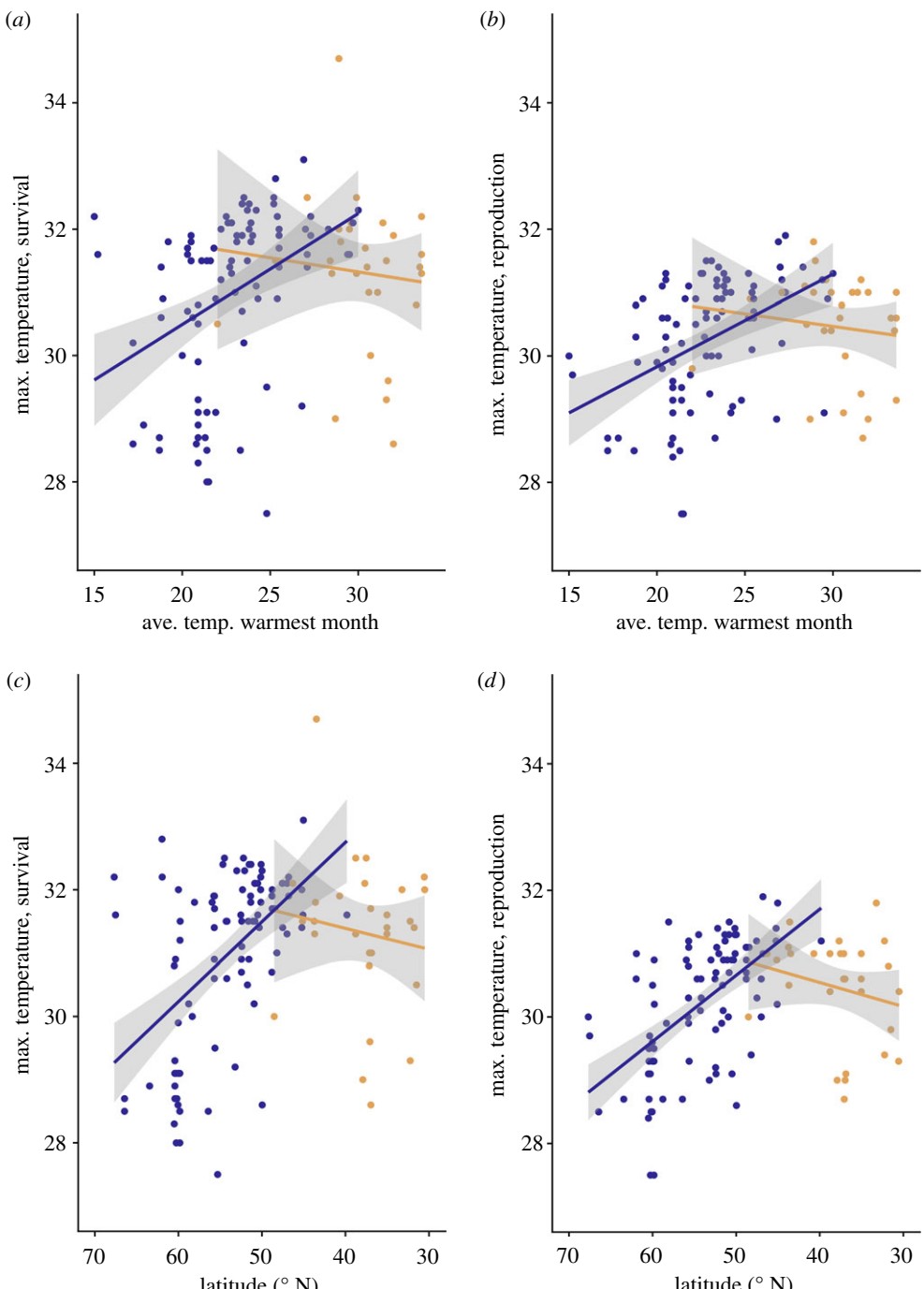

**Figure 3.** Scatter plot of maximal temperature for survival (a,c) and reproduction (b,d) plotted against the average temperature of the warmest month at the sampling site (a,b) and against latitude (c,d). Solid lines are fitted models of a linear regression. Blue symbols are summer-active populations (northern) and yellow symbols are winter-active (southern) populations. See table 1 for corresponding statistics. (Online version in colour.)

temperatures in their planktonic phase in summer (July/ August), while southern populations become active in winter and spring after the onset of the wet season. As predicted, we found a clear decline in temperature tolerance for summer-active populations with increasing latitude and decreasing temperature during the warmest month. For winter-active populations, this relationship was not found. For them, summer temperatures are not a limiting factor, as ponds are typically dry in summer and *Daphnia* undergo diapause. Thus, a high temperature tolerance is not expected for planktonic phase *Daphnia* from winter-active populations. Overall, our study confirms the prediction of local temperature adaptation for *D. magna*, and explains why earlier studies found mixed results [20,23–25].

Latitude and the average maximum temperature of the warmest month are shown here to be good predictors for upper temperature tolerance of summer-active populations. Over large geographical ranges and in the absence of strong altitudinal variation, these two variables are expected to correlate strongly. However, while the average temperature of the warmest month is a likely proximate driver of local temperature adaptation, latitude is the ultimate driver [31].

Local adaptation with regard to climatic variables is expected to change gradually across space, because sites located in close proximity are expected to have a similar climate. This is in contrast with habitat features that may vary strongly on small spatial scales (e.g. water quality and presence/absence of parasites and predators). To test this, we

**Table 1.** Analysis of covariance tables with latitude or average highest temperature of the warmest month (AHT_warmest) as continuous variable and summer_active as factor (0/1). A significant interaction term indicates a lack of homogeneity of slopes for summer- and winter-active populations. In (a) and (c), statistics for the maximal survival temperature are shown; in (b) and (d), statistics for maximal reproduction temperature are shown. (a) and (b) use AHT_warmest as covariable, (c) and (d) use latitude as covariable. See figure 3 for the corresponding figure.

| | d.f. | mean$^2$ | F-value | p (>F) |
|---|---|---|---|---|
| (a) model: maxtempsurv ~ summer_active × AHT_warmest | | | | |
| summer_active | 1 | 2.5518 | 1.5989 | 0.208528 |
| AHT_warmest | 1 | 18.9125 | 11.8501 | 0.000796*** |
| summer_active : AHT_warmest | 1 | 7.3794 | 4.6237 | 0.033556* |
| residuals | 119 | 1.5960 | | |
| (b) model: maxtempfec ~ summer_active × AHT_warmest | | | | |
| summer_active | 1 | 1.6195 | 1.7308 | 0.190835 |
| AHT_warmest | 1 | 11.1837 | 11.9523 | 0.000758*** |
| summer_active : AHT_warmest | 1 | 4.6962 | 5.0189 | 0.026928* |
| residuals | 119 | 0.9357 | | |
| (c) model: maxtempsurv ~ summer_active × latitude | | | | |
| summer_active | 1 | 2.552 | 1.8039 | 0.18180 |
| latitude | 1 | 33.925 | 23.9816 | $3.09 \times 10^{-6}$*** |
| summer_active : latitude | 1 | 13.946 | 9.8583 | 0.00213** |
| residuals | 119 | 1.415 | | |
| (d) model: maxtempfec ~ summer_active × Latitude | | | | |
| summer_active | 1 | 1.6195 | 2.0762 | 0.152239 |
| latitude | 1 | 22.9662 | 29.4423 | $3.08 \times 10^{-7}$*** |
| summer_active : latitude | 1 | 11.4362 | 14.6610 | 0.000207*** |
| residuals | 119 | 0.7800 | | |

Significance codes: ***$p < 0.001$, **$p < 0.01$, *$p < 0.05$.

used a spatial autocorrelation analysis. We found very high autocorrelations for the two estimators of tolerance used here (figure 4) strongly supporting our suggestion that temperature tolerance in *D. magna* evolved in response to the local climate. The analysis confirms that the high variation among *D. magna* genotypes observed here is closely linked to geography: up to a distance of about 1000 km, the closer two populations are located to each other, the more similar their heat tolerance.

### (a) Local adaptation in summer-active populations

Summer-active populations experience summer temperatures every year. Genotypes unable to tolerate the local temperatures are less likely to contribute to a population's genepool. Thus, higher temperatures are expected to select for a tolerance to higher temperatures. Experimental evolution and sediment core resurrection studies of *D. magna* support this interpretation [5].

Local adaptation is only found when the trait provides a specific benefit that would be disadvantageous in other locations [32], such as tolerance to high temperatures, which would be disadvantageous in colder locations. If a tolerance to high temperatures has costs, tolerance will only be selected for in climates where it is advantageous. At colder locations, selection would favour individuals that pay less costs, and these are those with a lower tolerance. Different trade-off models may apply to our findings [33]: first, by

evolving a higher temperature tolerance, organisms may have costs expressed as a generally reduced fitness. Second, by evolving a higher temperature tolerance, organisms may have costs in reduced tolerance to lower temperature. This explanation was suggested for temperature tolerance in ants [3]. Both of these hypotheses can be tested with further experiments and are consistent with diverse physiological explanations, such as homeoviscous adaptation, expression of chaperonins, shifts in osmolyte systems and expression of paralogue enzymes (reviewed in [34]).

### (b) Local adaptation in winter-active populations

The *Daphnia* genotypes collected in the species' subtropical range had large variation in heat tolerance, but that variation was not explained by summer temperatures or latitude. This does not prove that subtropical populations are not locally adapted to the ambient temperature conditions, but rather that the variables used here (local warmest summer temperature and latitude) are poorly correlated with the factor driving selection for temperature adaptation. Seasonality of precipitation in subtropical climates is likely to play a larger role because the reemergence of freshwater organisms from diapause is strongly linked to the onset of rainfall, which is usually during winter—the wettest and coldest period in the Holarctic subtropics [30]. However, because the rainfall distribution is highly variable in regions with winter-active populations, it is difficult to assess the warmest period with

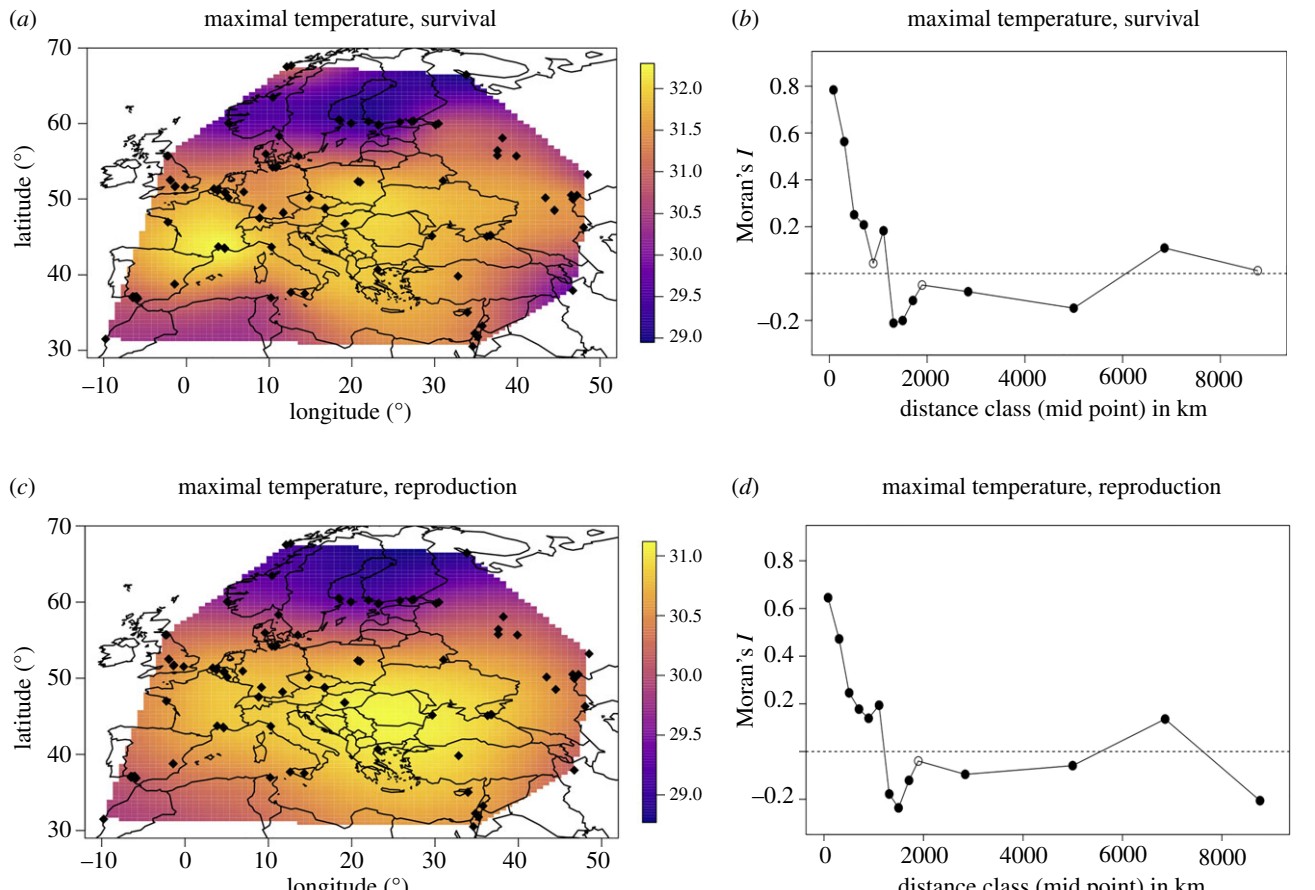

**Figure 4.** (*a,c*) Heat map of maximal temperature for survival (*a*) and reproduction (*c*) across the Western Palaearctic (−10° to 60° E). Small black dots mark sampling locations of *D. magna* genotypes. (*b,d*) Moran spatial autocorrelation (*I*) of maximal temperature for survival (*b*) and reproduction (*d*). Distance classes become larger with increasing distance. Filled symbols mark significant correlation coefficients ($p < 0.05$). (Online version in colour.)

suitable water conditions for winter-active populations. The beginning of the rainy season varies regionally and stochastically from early autumn to late winter (figure 1). In very dry locations, entire winters may pass without sufficient rain to support a *Daphnia* population. Furthermore, the length of the hydroperiod may differ among locations, lasting from a few weeks to several months. Therefore, the temperature of the warmest month with suitable water conditions is much harder to estimate for winter-active populations than for summer-active populations, making it difficult to correlate estimators of temperature with temperature tolerance for winter-active populations. Thus, while our results indicate that winter-active populations are not adapted to the temperature of the warmest month of the year, we have no reason to assume that they are not adapted to their local temperature conditions, or to other habitat features. For example, Roulin *et al.* [27] showed local adaptation of winter-active populations of *D. magna* regarding resting stages, which are predominantly produced when the photoperiod is increasing, indicating the end of the winter. Other traits (e.g. emergence from diapause) may be influenced by this habitat difference as well.

## (c) Spatial variation in genetic load and inbreeding

In our data, the *Daphnia* genotypes exhibiting the lowest temperature tolerance were mostly from northern rock pool populations (about north of 59°). These populations are part of highly dynamic metapopulations with frequent local

extinction and re-colonization events [35] that often give rise to population bottlenecks, and, consequently, to a high genetic load and high levels of inbreeding [26,36]. Recent studies speculate that these recurring events cause deleterious mutations to accumulate, and, combined with inbreeding, lead to reduced fitness [26,37]. Thus, the presumably poor genetic health of these northern genotypes might explain their low tolerance to high temperatures. This alternative hypothesis could be tested in future experiments: genetic crosses among rock-pool-derived clones from locations with the same temperature profile would reduce the effect of inbreeding, while keeping the expectation for local temperature adaptation the same (for a similar argument, see [37]).

On the other hand, local adaptation at the metapopulation level has been previously reported [27] and shows evidence for directional selection [38]. Nordic rock pool populations were found to have the highest resting egg production among European populations, reflecting an adaptation to the high instability of the rock pool habitat. Thus, the poor genetic health of rock pool *D. magna* seems not to hinder local adaptation.

## 5. Conclusion

Our study demonstrates that phenological variation within a species is an essential element influencing local temperature adaptation of populations across a wide geographical

range. We demonstrated this for the model system *D. magna*, but it also holds true for other organisms. For example, other cladoceran species (e.g. *Coronatella (Alona) rectangula*, *Ceriodaphnia quadrangula* and *Simocephalus vetulus*) and numerous insect species are temperature-limited at higher latitudes and precipitation-limited closer to the tropics [14–17,39–41]. More generally, our finding may apply to many species with a wide geographical range and a diapause phase that allows them to escape periods of the year with unsuitable conditions. As climate warming affects populations worldwide, populations may experience this change not only as an increase in temperature, but also with a change in phenology [12,42,43] or voltinism [17,44], which may, in some cases, even lead to counterintuitive relationships between latitude and the average temperature a population experiences during its active phase of the life cycle [11].

Data accessibility. All data are available in electronic supplementary material, table S1.

Authors' contributions. L.S. and D.E. conceived and designed the experiment. L.S. performed the experiment. L.S. and D.E. conducted the analyses and wrote the manuscript.

Competing interests. We declare we have no competing interests.

Funding. This study was supported by Swiss National Science Foundation (grant no. 310030B_166677).

Acknowledgements. We thank Yan Galimov and Frida Ben-Ami for help with collecting temperature records using data loggers in the field. We thank the following people for their help in providing *D. magna* genotypes for this study: Adam Petrusek, Alexey Kotov, Benjamin Lange, Christoph Haag, Daniela Brunner, Dieter Ebert, Elham Sheik-Jabbari, Ellen Decaestecker, Eric von Elert, Federico Morrone, France Dufresne, Frida Ben-Ami, Iakovos Tziortzis, Ioana Enache, Ivan Gomez-Mestre, Jarkko Routtu, Jason Andras, Juergen Hottinger, Kai Lyu, Kay van Damme, Knut-Helge Jensen, Lev Yampolsky, Liron Goren, Luc DeMeester, Lukas Schaerer, Meryem Beklioglu, Nadja Brun, Peter Fields, Raquel Ortells Baneres, Samuel Pichon, Sandra Lass, Sergey Glagolev, Sigurd Einum, Souad Turki, Sue Mitchell, Theodosiou Loukas, Thomas Zumbrunn, Tom Little, Vasil Vezhnavets, Andrei Papkou, Vladimir Tchougounov and Yan Galimov. We thank Jürgen Hottinger, Michelle Krebs and Urs Stiefel for assistance in the laboratory. Maridel Fredericksen, Jonathan Stillman and the members of the Ebert research group made helpful comments on the manuscript. Suzanne Zweizig improved the language of the text.

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
