## [Reviewer comments · Proceedings of the Royal Society B: Biological Sciences]

Review History

RSPB-2019-0929.R0 (Original submission)

Review form: Reviewer 1

Recommendation

Accept with minor revision (please list in comments)

Scientific importance: Is the manuscript an original and important contribution to its field?

Excellent

General interest: Is the paper of sufficient general interest?

Excellent

Quality of the paper: Is the overall quality of the paper suitable?

Excellent

Is the length of the paper justified?

Yes

Should the paper be seen by a specialist statistical reviewer?

No

Do you have any concerns about statistical analyses in this paper? If so, please specify them explicitly in your report.

No

It is a condition of publication that authors make their supporting data, code and materials available - either as supplementary material or hosted in an external repository. Please rate, if applicable, the supporting data on the following criteria.

Is it accessible?

Yes

Is it clear?

Yes

Is it adequate?

Yes

Do you have any ethical concerns with this paper?

No

Comments to the Author

Herewith, I'd like to submit my comments on the manuscript „Temperature- versus precipitation-limitation shape local temperature tolerance in a Holarctic freshwater crustacean“ by L. Seefeldt & D. Ebert.

—————
General comments:

The authors address an important question of local temperature adaptation using an ecological relevant aquatic model organism. In particular, they tested in how far phenology might be an additionally driving factor by investigating the upper temperature limits of an impressive number of *Daphnia* clones sampled from around the world from many different latitudes. They distinguished the clones into summer- and winter-active populations and linked the measured temperature tolerances to the respective potential local adaptation. While for winter-active populations there was no clear relationship, summer-active populations were locally adapted and temperature tolerances highly autocorrelated among neighbouring populations. The study concludes that in future phenology should be taken into account when trying to predict the influences of climate change.

In general, the manuscript is well written, with a nice introduction and fancy, enjoyable story line, mainly clearly described methods and adequately applied state of the arts statistical procedures. I certainly enjoyed reading this manuscript.

I have just a few minor comments on the method description, and data analysis, which to some parts appears a bit redundant, although a reasoning for the chosen approaches might be adequate to solve this already. I am sure that slight modifications of the writing might increase the accessibility for a potential readership of *Roy. Proc. Soc. B*. I am looking forward seeing this fine manuscript published.

—————
Minor comments:

Specifics on the Methods: „... We then took animals from these 28 °C cultures and tested for reproduction and survival at nine higher temperatures (29 to 37 °C). For this, we collected three juvenile females (1 -3 days old) from each of the 28 °C replicate populations and placed them in a 100-mL jar with 80 mL medium. These jars were kept in an incubator at the higher temperature (assay temperature). Animals were checked for survival and reproduction. If at least one animal of a replicate survived the 14-day assay period, the same replicate was tested at the next higher

temperature, but starting with juveniles taken from the 28 °C cultures. If none of the animals survived 14 days at a certain temperature, this replicate was still tested at the next higher temperature. If all animals died again, we assumed that we had exceeded its upper temperature tolerance and discontinued with this replicate.“

I am bit puzzled about which animals were chosen for the tests, although the supplementary figure does a good job. If all temperature assays were started with animals from 28°C then there should have been at least 10 beakers with 3 juveniles each per replicate at the beginning of the test. If I get it right, the temperature ramp was only used to save time and animals?

Hypothetically, everything could have been done in parallel given enough time and animals. This is a highly reasonable approach, and it's clear that most probably the complex design is difficult to describe; it's just to make sure that I got it right, or if there's another rationale behind. Perhaps adding a bit more information to the suppl. Fig. might help - or - telling more about the alternative rationale ;-)

— — — — —
Specifics on the data analysis / statistics:

While I understand why authors chose to analyse their data with a) separate regression for summer vs. winter active clones, b) ANOVA including both summer+winter-active clones, to show different slopes among these subsets, and finally c) the quadratic linear model, I have the feeling that presenting all three approaches in the main text might be a bit redundant. Perhaps one could describe the primary approach and the rationale behind the chosen approach in the methods, (the three analyses build up on each other) and add details and separate interpretations in the appendix? But this is clearly a matter of taste.

— — — — —
specific comments:

abstract: line 43: is „now“ needed?

intro:

line 109/110: I suggest to replace the 2nd „kept to temperatures“ by something like „tested from ... to ... „

— — — — —
methods:

line 147/148: give more details on the two diet species, not only their genus.

lines 200-203: probably, it's a nicer flow, if you first tell that you divided the data into three (spatial) categories with similar number of observations, and then mention the distance intervals chosen.

results:

using the direct variable names from R might be efficient, but not so nice for people not working with this software, I would suggest to use nice letters for the variables with subscripts, etc.

lines 220-225 more discussion than result, think of moving to discussion section!

lines 228 - 232: please give rationale behind interpretation of variance components, at the present I cannot nicely follow your argumentation line. A bit more details might be needed here.

line 234: I would avoid a rating of your results such as „not surprising“ .

lines 236 - 246 poly-fits assume an optimum curve here, summer and winter clones contribute to different parts of this optimum curve. You cannot really argue that the summer clones have a dependency but winter clones do not; the optimum curve may even suggest a decrease at higher x-axis values. Otherwise, one should/could have applied a saturation function. As you see, I have the feeling that the reader has to be guided a bit better through your ideas here and your rationale behind the analysis and presentation of data.

line 254: „adapted“ instead of „adaptation“

line 288 fig 4 not fig. 5

line 333 „winter-active populations are specifically adapted to their habitat“ ? I am not sure, if I mixed something up, but doesn't the „regression“ suggest the opposite?

lines 364-368: literature  not really a conclusion!

table 1: shows the interactions between (summer active/not active) x (latitude) are significant,

why you've chosen the poly(trait ~ latitude) model? As mentioned earlier, a reader might ask for the rationale behind.

a minor thing on the figures: colour codes: printed in grey scale: blue and green background is not distinguishable (at least on my printer), green-red was nicely distinguishable, but probably difficult for colour-blind people.

-end or review-

— — — — —

Review form: Reviewer 2

Recommendation

Accept with minor revision (please list in comments)

Scientific importance: Is the manuscript an original and important contribution to its field?

Good

General interest: Is the paper of sufficient general interest?

Excellent

Quality of the paper: Is the overall quality of the paper suitable?

Excellent

Is the length of the paper justified?

Yes

Should the paper be seen by a specialist statistical reviewer?

No

Do you have any concerns about statistical analyses in this paper? If so, please specify them explicitly in your report.

Yes

It is a condition of publication that authors make their supporting data, code and materials available - either as supplementary material or hosted in an external repository. Please rate, if applicable, the supporting data on the following criteria.

Is it accessible?

Yes

Is it clear?

Yes

Is it adequate?

Yes

Do you have any ethical concerns with this paper?

No

Comments to the Author

This manuscript addresses the general issue of how taxa with wide geographic distributions adapt locally. Over large distances the same organisms are subjected to different selective

regimes and this can hinder our ability to predict how populations will cope with locally changing conditions. Therefore, this manuscript specifically focusses upon local adaptation in temperature tolerance, taking into account seasonality and interactions with the organism's phenology. The authors used over 120 clones of the freshwater crustacean *Daphnia magna* collected over a latitudinal gradient and subjected them to increasing temperatures, measuring both survival and reproductive output. The experimental study was complemented with data logger and database data on temperature regimes at the sites where each of the genotypes originated. All the analyses were performed in R with various packages (note: the relevance of those packages is not evaluated in this review).

A major conclusion is that genetics considerably contribute to temperature tolerance traits, as tolerance increased with the average highest temperature at the site of origin for the clones. However, this effect was only found in populations that were active in summer (over 43°N) whereas winter-active populations are dormant and therefore not limited by temperature. Therefore, it is important to consider the phenology of the organism and the seasonality of the local environment.

Major comments

- Further explanation is required with regards to the rationale for using spatial autocorrelation analysis in the materials and methods section. Currently it reads as a very interesting tool with little rationale and minimal interpretation. Also, further detail is needed regarding how the spatial analysis is conducted as it has distance classes up to 8000km which would include nearly the entire Northern hemisphere, yet it is combined into a figure looking only at the Western Palearctic, so it is misleading.
- Moran's I was, as I understand, used to detect spatial autocorrelation among locations in two dimensions, i.e. including both latitudinal and longitudinal positions. However, when addressing temperature there is a steep gradient along latitudes, whereas the longitudinal gradient is negligible. This would suggest that the autocorrelation in survival and reproductive responses should be very strong among sites along the longitudinal gradient, whereas it would be expected to be strong at short distances and weak over long distances latitudinally. Hence, is Moran's I really the best way to test this? Why not use a much simpler one-dimensional autocorrelation and provide figures for both latitudinal and longitudinal autocorrelations? This would, in addition, be a critical test of if temperature is the driving force, i.e. testing the prediction that there is strong longitudinal, but no latitudinal autocorrelation among sites.
- The 123 clones were retrieved from a clone collection. How long had the different clones been cultivated and under what conditions? Please clarify and discuss if clones may have adapted to lab conditions before being included in the experiment.

Minor comments

Line.

36. consider changing 'seasonal phenology' to just 'phenology' or 'the phenology of the organism' as seasonal is implicit

36. use lower case l in latitude

42. 'and summer' □ and local summer

58. Turn into one sentence i.e. – To predict how...temperature conditions, it is necessary to...

83. 'Rather, the seasonal phenology and diapause need to be taken into account' these things are essentially the same so consider changing to something like □ Therefore, the phenology of the organism needs to be taken into account.

85. 'seasonal phenology, diapause and local temperature adaptation' □ phenology and local

temperature adaptation

92. Italicise *Daphnia*

110. 'kept to' □ kept at

115. Climate warming will come with other environmental changes than temperature increase (for example increase rainfall in certain regions) so consider changing this sentence to 'We conclude that only in summer-active populations do summer temperatures limit biological function, thus climate warming may directly influence patterns of local adaptation in these populations'.

207. consider swapping the first and second sentences

216. Italicise *Daphnia*

233. not clear at what temperatures *Daphnia* were kept, in the introduction it was 36, materials it was 37 and in results its 35

234. Not surprising = Not surprisingly

249. -10 and 60 degrees E

254. 'adaptation' □ adapted

265. remove '(non-dormant)'

279. '*Daphnia magna*' □ *D. Magna*

288. Figure 5 does not exist, at least not in the copy I got. There are only four figures in the manuscript. Please correct.

348. 'low temperature tolerance' □ low tolerance to high temperatures

570. 'Dezember' □ December

582 & 584. Use the same format 'Left: & (right)' for example '(Left) & (Right)'

Decision letter (RSPB-2019-0929.R0)

04-Jun-2019

Dear Dr Ebert:

Your manuscript has now been peer reviewed and the reviews have been assessed by an Associate Editor. The reviewers' comments (not including confidential comments to the Editor) and the comments from the Associate Editor are included at the end of this email for your reference. As you will see, the reviewers and the Editors have raised some concerns with your manuscript and we would like to invite you to revise your manuscript to address them.

When submitting your revision please upload a file under "Response to Referees" - in the "File Upload" section. This should document, point by point, how you have responded to the reviewers' and Editors' comments, and the adjustments you have made to the manuscript. We

require a copy of the manuscript with revisions made since the previous version marked as 'tracked changes' to be included in the 'response to referees' document.

Research ethics:

Use of animals and field studies:

If you wish to submit your data to Dryad (<http://datadryad.org/>) and have not already done so you can submit your data via this link [http://datadryad.org/submit?journalID=RSPB&manu=\(Document not available\)](http://datadryad.org/submit?journalID=RSPB&manu=(Document%20not%20available)), which will take you to your unique entry in the Dryad repository.

Online supplementary material will also carry the title and description provided during submission, so please ensure these are accurate and informative. Note that the Royal Society will

not edit or typeset supplementary material and it will be hosted as provided. Please ensure that the supplementary material includes the paper details (authors, title, journal name, article DOI). Your article DOI will be 10.1098/rspb.[paper ID in form xxxx.xxxx e.g. 10.1098/rspb.2016.0049].

Please submit a copy of your revised paper within three weeks. If we do not hear from you within this time your manuscript will be rejected. If you are unable to meet this deadline please let us know as soon as possible, as we may be able to grant a short extension.

Best wishes,
Dr Daniel Costa
mailto:proceedingsb@royalsociety.org

Associate Editor
Board Member: 1
Comments to Author:
Dear Dieter Ebert

Your manuscript has been reviewed by two reviewers and they both agree that your paper should be publishable pending minor revisions. They provide thorough and considerate suggestions as to the points this revision should cover. Please submit a point by point revision and cover letter and I will make a final recommendation on your manuscript.

Best

Line K Bay

Reviewer(s)' Comments to Author:

Referee: 1

Comments to the Author(s)

Herewith, I'd like to submit my comments on the manuscript „Temperature- versus precipitation-limitation shape local temperature tolerance in a Holarctic freshwater crustacean“ by L. Seefeldt & D. Ebert.

— — — — —
General comments:

The authors address an important question of local temperature adaptation using an ecological relevant aquatic model organism. In particular, they tested in how far phenology might be an additionally driving factor by investigating the upper temperature limits of an impressive number of *Daphnia* clones sampled from around the world from many different latitudes. They distinguished the clones into summer- and winter-active populations and linked the measured temperature tolerances to the respective potential local adaptation. While for winter-active populations there was no clear relationship, summer-active populations were locally adapted and temperature tolerances highly autocorrelated among neighbouring populations. The study concludes that in future phenology should be taken into account when trying to predict the influences of climate change.

In general, the manuscript is well written, with a nice introduction and fancy, enjoyable story line, mainly clearly described methods and adequately applied state of the arts statistical procedures. I certainly enjoyed reading this manuscript.

I have just a few minor comments on the method description, and data analysis, which to some parts appears a bit redundant, although a reasoning for the chosen approaches might be adequate to solve this already. I am sure that slight modifications of the writing might increase the accessibility for a potential readership of Roy. Proc. Soc. B. I am looking forward seeing this fine manuscript published.

— — — — —
 Minor comments:

Specifics on the Methods: „... We then took animals from these 28 °C cultures and tested for reproduction and survival at nine higher temperatures (29 to 37 °C). For this, we collected three juvenile females (1 -3 days old) from each of the 28 °C replicate populations and placed them in a 100-mL jar with 80 mL medium. These jars were kept in an incubator at the higher temperature (assay temperature). Animals were checked for survival and reproduction. If at least one animal of a replicate survived the 14-day assay period, the same replicate was tested at the next higher temperature, but starting with juveniles taken from the 28 °C cultures. If none of the animals survived 14 days at a certain temperature, this replicate was still tested at the next higher temperature. If all animals died again, we assumed that we had exceeded its upper temperature tolerance and discontinued with this replicate.“

I am bit puzzled about which animals were chosen for the tests, although the supplementary figure does a good job. If all temperature assays were started with animals from 28°C then there should have been at least 10 beakers with 3 juveniles each per replicate at the beginning of the test. If I get it right, the temperature ramp was only used to save time and animals?

Hypothetically, everything could have been done in parallel given enough time and animals. This is a highly reasonable approach, and it's clear that most probably the complex design is difficult to describe; it's just to make sure that I got it right, or if there's another rationale behind. Perhaps adding a bit more information to the suppl. Fig. might help - or - telling more about the alternative rationale ;-)

— — — — —
 Specifics on the data analysis / statistics:

While I understand why authors chose to analyse their data with a) separate regression for summer vs. winter active clones, b) ANOVA including both summer+winter-active clones, to show different slopes among these subsets, and finally c) the quadratic linear model, I have the feeling that presenting all three approaches in the main text might be a bit redundant. Perhaps one could describe the primary approach and the rationale behind the chosen approach in the methods, (the three analyses build up on each other) and add details and separate interpretations in the appendix? But this is clearly a matter of taste.

— — — — —
 specific comments:

abstract: line 43: is „now“ needed?

intro:

line 109/110: I suggest to replace the 2nd „kept to temperatures“ by something like „tested from to ... „

— — — — —
 methods:

line 147/148: give more details on the two diet species, not only their genus.

lines 200-203: probably, it's a nicer flow, if you first tell that you divided the data into three (spatial) categories with similar number of observations, and then mention the distance intervals chosen.

results:

using the direct variable names from R might be efficient, but not so nice for people not working with this software, I would suggest to use nice letters for the variables with subscripts, etc.

lines 220-225 more discussion than result, think of moving to discussion section!

lines 228 – 232: please give rationale behind interpretation of variance components, at the present I cannot nicely follow your argumentation line. A bit more details might be needed here.

line 234: I would avoid a rating of your results such as „not surprising“ .

lines 236 – 246 poly-fits assume an optimum curve here, summer and winter clones contribute to different parts of this optimum curve. You cannot really argue that the summer clones have a dependency but winter clones do not; the optimum curve may even suggest a decrease at higher x-axis values. Otherwise, one should/could have applied a saturation function. As you see, I have the feeling that the reader has to be guided a bit better through your ideas here and your rationale behind the analysis and presentation of data.

line 254: „adapted“ instead of „adaptation“

line 288 fig 4 not fig. 5

line 333 „winter-active populations are specifically adapted to their habitat“ ? I am not sure, if I mixed something up, but doesn't the „regression“ suggest the opposite?

lines 364-368: literature  not really a conclusion!

table 1: shows the interactions between (summer active/not active) x (latitude) are significant, why you've chosen the poly(trait ~ latitude) model? As mentioned earlier, a reader might ask for the rationale behind.

a minor thing on the figures: colour codes: printed in grey scale: blue and green background is not distinguishable (at least on my printer), green-red was nicely distinguishable, but probably difficult for colour-blind people.

-end or review-

Referee: 2

Comments to the Author(s)

This manuscript addresses the general issue of how taxa with wide geographic distributions adapt locally. Over large distances the same organisms are subjected to different selective regimes and this can hinder our ability to predict how populations will cope with locally changing conditions. Therefore, this manuscript specifically focusses upon local adaptation in temperature tolerance, taking into account seasonality and interactions with the organism's phenology. The authors used over 120 clones of the freshwater crustacean *Daphnia magna* collected over a latitudinal gradient and subjected them to increasing temperatures, measuring both survival and reproductive output. The experimental study was complemented with data logger and database data on temperature regimes at the sites where each of the genotypes originated. All the analyses were performed in R with various packages (note: the relevance of those packages is not evaluated in this review).

A major conclusion is that genetics considerably contribute to temperature tolerance traits, as tolerance increased with the average highest temperature at the site of origin for the clones. However, this effect was only found in populations that were active in summer (over 43°C) whereas winter-active populations are dormant and therefore not limited by temperature. Therefore, it is important to consider the phenology of the organism and the seasonality of the local environment.

Major comments

- Further explanation is required with regards to the rationale for using spatial autocorrelation analysis in the materials and methods section. Currently it reads as a very interesting tool with

little rationale and minimal interpretation. Also, further detail is needed regarding how the spatial analysis is conducted as it has distance classes up to 8000km which would include nearly the entire Northern hemisphere, yet it is combined into a figure looking only at the Western Palearctic, so it is misleading.

- Moran's I was, as I understand, used to detect spatial autocorrelation among locations in two dimensions, i.e. including both latitudinal and longitudinal positions. However, when addressing temperature there is a steep gradient along latitudes, whereas the longitudinal gradient is negligible. This would suggest that the autocorrelation in survival and reproductive responses should be very strong among sites along the longitudinal gradient, whereas it would be expected to be strong at short distances and weak over long distances latitudinally. Hence, is Moran's I really the best way to test this? Why not use a much simpler one-dimensional autocorrelation and provide figures for both latitudinal and longitudinal autocorrelations? This would, in addition, be a critical test of if temperature is the driving force, i.e. testing the prediction that there is strong longitudinal, but no latitudinal autocorrelation among sites.
- The 123 clones were retrieved from a clone collection. How long had the different clones been cultivated and under what conditions? Please clarify and discuss if clones may have adapted to lab conditions before being included in the experiment.

Minor comments

Line.

36. consider changing 'seasonal phenology' to just 'phenology' or 'the phenology of the organism' as seasonal is implicit

36. use lower case l in latitude

42. 'and summer' □ and local summer

58. Turn into one sentence i.e. – To predict how...temperature conditions, it is necessary to...

83. 'Rather, the seasonal phenology and diapause need to be taken into account' these things are essentially the same so consider changing to something like □ Therefore, the phenology of the organism needs to be taken into account.

85. 'seasonal phenology, diapause and local temperature adaptation' □ phenology and local temperature adaptation

92. Italicise *Daphnia*

110. 'kept to' □ kept at

115. Climate warming will come with other environmental changes than temperature increase (for example increase rainfall in certain regions) so consider changing this sentence to 'We conclude that only in summer-active populations do summer temperatures limit biological function, thus climate warming may directly influence patterns of local adaptation in these populations'.

207. consider swapping the first and second sentences

216. Italicise *Daphnia*

233. not clear at what temperatures *Daphnia* were kept, in the introduction it was 36, materials it was 37 and in results its 35

234. Not surprising = Not surprisingly

249. -10 and 60 degrees E

254. 'adaptation' □ adapted

265. remove '(non-dormant)'

279. '*Daphnia magna*' □ *D. Magna*

288. Figure 5 does not exist, at least not in the copy I got. There are only four figures in the manuscript. Please correct.

348. 'low temperature tolerance' □ low tolerance to high temperatures

570. 'Dezember' □ December

582 & 584. Use the same format 'Left: & (right)' for example '(Left) & (Right)'

Author's Response to Decision Letter for (RSPB-2019-0929.R0)

See Appendix A.

Decision letter (RSPB-2019-0929.R1)

01-Jul-2019

Dear Dr Ebert

I am pleased to inform you that your manuscript entitled "Temperature- versus precipitation-limitation shape local temperature tolerance in a Holarctic freshwater crustacean" has been accepted for publication in Proceedings B.

Open Access

Your article has been estimated as being 9 pages long. Our Production Office will be able to confirm the exact length at proof stage.

Paper charges

All supplementary materials accompanying an accepted article will be treated as in their final form. They will be published alongside the paper on the journal website and posted on the online

figshare repository. Files on figshare will be made available approximately one week before the accompanying article so that the supplementary material can be attributed a unique DOI.

Sincerely,

Dr Daniel Costa
Editor, Proceedings B
mailto: proceedingsb@royalsociety.org

Associate Editor:
Board Member
Comments to Author:
(There are no comments.)

Appendix A

Response to Referees

Reviewer(s)' Comments to Author:

Referee: 1

Comments to the Author(s)

Herewith, I'd like to submit my comments on the manuscript „Temperature- versus precipitation-limitation shape local temperature tolerance in a Holarctic freshwater crustacean“ by L. Seefeldt & D. Ebert.

— — — — —
General comments:

The authors address an important question of local temperature adaptation using an ecological relevant aquatic model organism. In particular, they tested in how far phenology might be an additionally driving factor by investigating the upper temperature limits of an impressive number of *Daphnia* clones sampled from around the world from many different latitudes. They distinguished the clones into summer- and winter-active populations and linked the measured temperature tolerances to the respective potential local adaptation. While for winter-active populations there was no clear relationship, summer-active populations were locally adapted and temperature tolerances highly autocorrelated among neighbouring populations. The study concludes that in future phenology should be taken into account when trying to predict the influences of climate change.

In general, the manuscript is well written, with a nice introduction and fancy, enjoyable story line, mainly clearly described methods and adequately applied state of the arts statistical procedures. I certainly enjoyed reading this manuscript.

I have just a few minor comments on the method description, and data analysis, which to some parts appears a bit redundant, although a reasoning for the chosen approaches might be adequate to solve this already. I am sure that slight modifications of the writing might increase the accessibility for a potential readership of Roy. Proc. Soc. B. I am looking forward seeing this fine manuscript published.

We are happy to read that the review liked our study. We will try to revise the manuscript following the detailed and useful suggestions.

— — — — —
Minor comments:

Specifics on the Methods: „... We then took animals from these 28 °C cultures and tested for reproduction and survival at nine higher temperatures (29 to 37 °C). For this, we collected three juvenile females (1 –3 days old) from each of the 28 °C replicate populations and placed them in a 100-mL jar with 80 mL medium. These jars were kept in an incubator at the higher temperature (assay temperature). Animals were checked for survival and reproduction. If at least one animal of a replicate survived the 14-day assay period, the same replicate was tested at the next higher temperature, but starting with juveniles taken from the 28 °C cultures. If none of the animals survived 14 days at a certain temperature, this replicate was still tested at the next higher temperature. If all animals died again, we assumed that we had exceeded its upper temperature tolerance and discontinued with this replicate.“

I am bit puzzled about which animals were chosen for the tests, although the supplementary figure does a good job. If all temperature assays were started with animals from 28°C then there should have been at least 10 beakers with 3 juveniles each per replicate at the beginning of the test. If I get it right, the temperature ramp was only used to save time and animals? Hypothetically, everything could have been done in parallel given enough time and animals. This is a highly reasonable approach, and it's clear that most probably the complex design is difficult to describe; it's just to make sure that I got it right, or if there's another rationale behind. Perhaps adding a bit more information to the suppl. Fig. might help – or – telling more about the alternative rationale

Indeed, after the animals had reached 28 °C, it would have been possible to run all tests at the same time. However, we had only 2 large walk-in climate chambers, one was set to 28 °C the other to 29, 30, 31, ... °C. We revised the text in the manuscript and in the suppl. file to make the procedure more clear.

Specifics on the data analysis / statistics:

While I understand why authors chose to analyse their data with a) separate regression for summer vs. winter active clones, b) ANOVA including both summer+winter-active clones, to show different slopes among these subsets, and finally c) the quadratic linear model, I have the feeling that presenting all three approaches in the main text might be a bit redundant. Perhaps one could describe the primary approach and the rationale behind the chosen approach in the methods, (the three analyses build up on each other) and add details and separate interpretations in the appendix? But this is clearly a matter of taste.

We understand the confusion created here. We changed this now. The ANCOVA fits best the message and we use now only this analysis. We also changed Fig. 3 accordingly). The other statistics are taken out. The text is much clearer now.

specific comments:

abstract: line 43: is „now“ needed?

We changed this.

intro:

line 109/110: I suggest to replace the 2nd „kept to temperatures“ by something like „tested from to ... „

We followed this suggestion.

methods:

line 147/148: give more details on the two diet species, not only their genus.

The algae are called: *Nanochloropsis limnetica* and *Acutodesmus obliquus*. *This is now included in the revised manuscript.*

lines 200-203: probably, it's a nicer flow, if you first tell that you divided the data into three (spatial) categories with similar number of observations, and then mention the distance intervals chosen.

We followed this suggestion.

results:

using the direct variable names from R might be efficient, but not so nice for people not working with this software, I would suggest to use nice letters for the variables with subscripts, etc.

We followed this suggestion.

lines 220-225 more discussion than result, think of moving to discussion section!

We left these two sentences at this place (rather than moving them to the discussion), as they give a clear description of the findings. For the reader it is important to understand this result at this place in the text, as it is essential for the analysis in the following paragraphs. We left it therefore as it is.

lines 228 – 232: please give rationale behind interpretation of variance components, at the present I cannot nicely follow your argumentation line. A bit more details might be needed here.

The point is to show that most of the overall variation observed is due to genetic effects. This is in line with the argument that the observed variation may be explained with local adaptation. We rephrased this paragraph to make this more clear.

line 234: I would avoid a rating of your results such as „not surprising“ .

We followed this suggestion.

lines 236 – 246 poly-fits assume an optimum curve here, summer and winter clones contribute to different parts of this optimum curve. You cannot really argue that the summer clones have a dependency but winter clones do not; the optimum curve may even suggest a decrease at higher x-axis values. Otherwise, one should/could have applied a saturation function. As you see, I have the feeling that the reader has to be guided a bit

better through your ideas here and your rationale behind the analysis and presentation of data.

We never used the word "optimum" (or anything similar) and we do not interpret the curve shown in the way implied by the referee. I guess our wording was misleading here. We try to improve this in the revised text and took the polynomial regression out of the text and the figure. The ANCOVA in Table 1 makes this point clear. We adapted the text and made Fig.3 and the table legend more clear now.

line 254: „adapted“ instead of „adaptation“

We followed this suggestion.

line 288 fig 4 not fig. 5

corrected

line 333 „winter-active populations are specifically adapted to their habitat“ ? I am not sure, if I mixed something up, but doesn't the „regression“ suggest the opposite?

*This was badly written by us. We changed the text to make clear what we intended to say. The new text reads now: " Thus, while our results indicate that winter-active populations are not adapted to the temperature of the warmest month of the year, we have no reason to assume that they are not adapted to their local temperature conditions, or to other habitat features. For example, Roulin et al. (2013) showed local adaptation of winter-active populations of *D. magna* regarding resting stages, which are predominantly produced when the photoperiod is increasing, indicating the end of the winter. "*

lines 364-368: literature  not really a conclusion!

We agree that taken in isolation, this sentence does not look like a conclusion. But these examples highlight that our results are likely valid for many other study systems. We therefore placed these examples in the conclusion to make the point of a wider implication of our results.

table 1: shows the interactions between (summer active/not active) x (latitude) are significant, why you've chosen the poly(trait ~ latitude) model? As mentioned earlier, a reader might ask for the rationale behind.

We changed this now. The ANCOVA in Table 1 is the proper analysis to work out the difference between summer-active and winter-active populations. We took the polynomial regression out and changed Fig. 3 accordingly. The polynomial regression does not really add anything new, but only shows the same in a different way.

a minor thing on the figures: colour codes: printed in grey scale: blue and green background is not distinguishable (at least on my printer), green-red was nicely distinguishable, but probably difficult for colour-blind people.

We changed the colour scheme. We use now the plasma colour scheme (part of R package "viridis") that solves this problem and also makes the figure well accessible for people with colour blindness.

-end or review-

Referee: 2

Comments to the Author(s)

This manuscript addresses the general issue of how taxa with wide geographic distributions adapt locally. Over large distances the same organisms are subjected to different selective regimes and this can hinder our ability to predict how populations will cope with locally changing conditions. Therefore, this manuscript specifically focusses upon local adaptation in temperature tolerance, taking into account seasonality and interactions with the organism's phenology. The authors used over 120 clones of the freshwater crustacean *Daphnia magna* collected over a latitudinal gradient and subjected them to increasing temperatures, measuring both survival and reproductive output. The experimental study was complemented with data logger and database data on temperature regimes at the sites where each of the genotypes originated. All the analyses were performed in R with various packages (note: the relevance of those packages is not evaluated in this review).

A major conclusion is that genetics considerably contribute to temperature tolerance traits, as tolerance increased with the average highest temperature at the site of origin for the clones. However, this effect was only found in populations that were active in summer (over 43°N) whereas winter-active populations are dormant and therefore not limited by temperature. Therefore, it is important to consider the phenology of the organism and the seasonality of the local environment.

Major comments

- Further explanation is required with regards to the rationale for using spatial autocorrelation analysis in the materials and methods section. Currently it reads as a very interesting tool with little rationale and minimal interpretation. Also, further detail is needed regarding how the spatial analysis is conducted as it has distance classes up to 8000km which would include nearly the entire Northern hemisphere, yet it is combined into a figure looking only at the Western Palearctic, so it is misleading.

We used spatial autocorrelation to show that the variation we uncover is indeed largely spatially correlated over shorter distances. Sample sites in close proximity are more similar to each other than sites far apart. This is what would be expected if climate is a driver behind the observed differences. Regarding the largest distance classes: We follow the way this is

used in the field. The expectation is that with large distances, the correlation should tend towards zero, which it does. Introducing a cut-off distance would require also a justification. The point is not to estimate the correlation for large distances as such, but rather to show that it goes down to zero and stays there. We explain this now better in the revised text of the discussion.

- Moran's I was, as I understand, used to detect spatial autocorrelation among locations in two dimensions, i.e. including both latitudinal and longitudinal positions. However, when addressing temperature there is a steep gradient along latitudes, whereas the longitudinal gradient is negligible. This would suggest that the autocorrelation in survival and reproductive responses should be very strong among sites along the longitudinal gradient, whereas it would be expected to be strong at short distances and weak over long distances latitudinally. Hence, is Moran's I really the best way to test this? Why not use a much simpler one-dimensional autocorrelation and provide figures for both latitudinal and longitudinal autocorrelations? This would, in addition, be a critical test of if temperature is the driving force, i.e. testing the prediction that there is strong longitudinal, but no latitudinal autocorrelation among sites.

This is a good point. We had already looked into the data to work this out. These analyses made it clear, that the intuition of us (and the reviewer here) is not justified (or at least not fully justified). Over the large range of sampling sites used here the intuition (no variation East-West, but a lot of change North-South) is correct, but across the distances where we find significant autocorrelation (up to about 800 km) this intuition does not help. On small local scales other factors play a role, such as continentality (coastal areas are more climatically buffered), altitude (colder at higher latitude), size of water body (larger bodies heat up less) and idiosyncratic factors, such as the role of the gulf stream. The isoclines for these factors are not going North-South. Thus, we decided to leave this aspect out of the analysis and kept the analysis simple. The point we want to make comes across very well with the use of Moran's I, so we decided to leave it as it is now presented in the manuscript.

- The 123 clones were retrieved from a clone collection. How long had the different clones been cultivated and under what conditions? Please clarify and discuss if clones may have adapted to lab conditions before being included in the experiment.

*Most clones used were collected between 2010 and 2014. A few clones were however older (up to 25 years). We do not believe that this plays an important role in the here presented experimental assessment of thermal tolerance. Populations kept in clonal stock in our lab are rather small and therefore adaptive evolutionary change is not very likely (genetic drift may happen, but would increase the noise, rather than the signal). We tested recently (Dukic et al. JEB 2019; doi: 10.1111/jeb.13443) if *D. magna* clones evolve by loss-of-heterozygosity (a mechanism proposed to contribute to evolution of clonal lines in the lab) and reached the conclusion that the impact of this is undetectable. Thus, all in all, we believe that our results are not (or hardly) influenced by the evolution in the laboratory. If this would be the case, it would likely reduce the signature of local adaptation, as all clones would adapt to the lab environment.*

Minor comments

Line.

36. consider changing 'seasonal phenology' to just 'phenology' or 'the phenology of the organism' as seasonal is implicit

We changed 'seasonal phenology' to just 'phenology' throughout the text.

36. use lower case l in latitude

changed

42. 'and summer' ◇ and local summer

changed

58. Turn into one sentence i.e. – To predict how...temperature conditions, it is necessary to...

changed

83. 'Rather, the seasonal phenology and diapause need to be taken into account' these things are essentially the same so consider changing to something like ◇ Therefore, the phenology of the organism needs to be taken into account.

changed

85. 'seasonal phenology, diapause and local temperature adaptation' ◇ phenology and local temperature adaptation

changed

92. Italicise Daphnia

changed

110. 'kept to' ◇ kept at

changed

115. Climate warming will come with other environmental changes than temperature increase (for example increase rainfall in certain regions) so consider changing this sentence to 'We conclude that only in summer-active populations do summer temperatures limit

biological function, thus climate warming may directly influence patterns of local adaptation in these populations’.

changed

207. consider swapping the first and second sentences

done

216. Italicise Daphnia

done

233. not clear at what temperatures Daphnia were kept, in the introduction it was 36, materials it was 37 and in results its 35

Sorry, this was confusing indeed and at one place (36 C) it was wrong (typo). As four replicates of one clone survived 35 °C, we stopped the experiment after testing these replicates at 36 and 37 °C. This is now made clear. No animal survived 35 °C.

234. Not surprising = Not surprisingly

We took this out in revising this section based on the other reviewers suggestions.

249. -10 and 60 degrees E

changed

254. ‘adaptation’ ◇ adapted

changed

265. remove ‘(non-dormant)’

done

279. ‘Daphnia magna’ ◇ D. Magna

changed

288. Figure 5 does not exist, at least not in the copy I got. There are only four figures in the manuscript. Please correct.

This was meant to be fig. 4. We corrected this.

348. ‘low temperature tolerance’ ◇ low tolerance to high temperatures

changed

570. 'Dezember' ◇ December

changed

582 & 584. Use the same format 'Left: & (right)' for example '(Left) & (Right)'

changed